# Assessing Soil Organic Carbon Stocks and Particle-Size Fractions across Cropping Systems in the Kiti Sub-Watershed in Central Benin

Arcadius Martinien Agassin Ahogle [1,2,*], Felix Kouelo Alladassi [1], Tobi Moriaque Akplo [1], Hessou Anastase Azontonde [3] and Pascal Houngnandan [1,4]

[1] Laboratoire de Microbiologie des Sols et d'Ecologie Microbienne, Faculté des Sciences Agronomiques, Université d'Abomey-Calavi, Abomey-Calavi BP 711, Benin
[2] Department of Spatial and Environmental Planning, Kenyatta University, Nairobi P.O. Box 43844-00100, Kenya
[3] Institut National des Recherches Agricoles du Bénin, CRA-Agonkanmey, Cotonou BP 884, Benin
[4] Ecole de Gestion et de Production Végétale et Semencière, Université Nationale d'Agriculture, Ketou BP 43, Benin
* Correspondence: ahoglearcadius@gmail.com

**Abstract:** Soil organic carbon storage in agricultural soil constitutes a crucial potential for sustainable agricultural productivity and climate change mitigation. This paper aimed at assessing soil organic carbon stock and its distribution in three particle size fractions across five cropping systems located in Kiti sub-watershed in Benin. Soil samples were collected using a grid sampling method on four soil depth layers: 0–10, 10–20, 20–30 and 30–40 cm in five cropping systems maize–cotton relay cropping (MCRC), yam–maize intercropping (YMI), teak plantation (TP), 5-year fallow (5YF) and above 10-year fallow (Ab10YF) from July to August 2017. Soil organic carbon stock (C stock) was estimated for the different soil layers and particle-size fractionation of soil organic matter was performed considering three fractions. The fractions coarse particulate organic matter (cPOM: 250–2000 μm), fine particulate organic matter (fPOM: 53–250 μm) and non-particulate organic matter (NOM: <53 μm) were separated from two soil depth layers: 0–10 and 10–20 cm. The results showed that fallow lands Ab10YF and 5YF exhibited the highest C stock, 22.20 and 17.74 Mg C·ha$^{-1}$, while cultivated land under tillage MCRC depicted the lowest, C stock 11.48 Mg C·ha$^{-1}$. The three organic carbon fractions showed a significant variation across the cropping systems with the NOM fraction holding the largest contribution to total soil organic carbon for all the cropping systems, ranging between 3.40 and 7.99 g/kg. The cPOM and fPOM were the most influenced by cropping systems with the highest concentration observed in Ab10YF and 5YF. The findings provide insights for upscaling farm management practices towards sustainable agricultural systems with substantial potential for carbon sequestration and climate change mitigation.

**Keywords:** carbon sequestration; sustainable farming systems; particulate organic carbon; particle-size fractionation

## 1. Introduction

Feeding an ever-growing population, expected to pass nine billion by 2050, in the tight context of climate change and resource degradation sets the greatest challenge for agricultural systems in the world [1]. Food and nutrition security is challenging in sub-Saharan Africa (SSA) where agricultural production is mainly rain-fed, relying mostly on traditional modes of farming with low input, i.e., most farmers do not use improved seed varieties and irrigation systems, apply low-rate organic amendment and mineral fertilizers, export crop residues and do not implement mechanized farming operations [2,3]. This type of farming system often leads to soil fertility depletion and disruption of its biochemical processes [4]. Along with the continuous exploitation of soil without replenishment, more

pressure has been imposed on the soil to cope with the ever-increasing food demand [5]. This has resulted in increasing soil degradation and subsequently a decrease in agricultural production [6].

Soils are crucial resources for agricultural production in that they help in water filtering, biodiversity preservation, atmospheric carbon storage and host biogeochemical processes [7,8]. As an essential reservoir of atmospheric carbon, soils have a vital role in the mitigation of greenhouse gas emissions [9]. Soil management is vital in environmental sustainability and achievement of the sustainable development goals (SDGs).

In SSA, soil fertility decline is a significant constraint hindering agricultural production [10]. In Benin, a country located in West Africa, about 70% of the total arable lands has been classified as low to very-low fertility [11]. This low soil fertility level is partially attributed to the intrinsic properties of these soils, i.e., low soil organic carbon (SOC) content and low-cation-exchange capacity [12]. In addition, poor farming practices, such as burning, crop residues exportation and nutrient miming contribute further to soil fertility depletion [4,12]. Indeed, soil organic carbon constitutes a key component of soil fertility and agroecosystem sustainability [13]. The literature has shown that the high content of soil organic matter in the surface layer is significantly correlated with a lower susceptibility to water erosion in West Africa [14–17]. According to Paul et al. [18], soil aggregation and soil structure stability increase with soil organic carbon content. Similarly, soil fauna diversity and activity are directly related to soil organic carbon content [19]. Furthermore, soil organic carbon influences fertilizer efficiency in agricultural production. However, different pools of SOC are involved in these processes [20]. The labile pool (<53–2000 μm) which has a few days to months turnover stimulates microbial activity and nutrient cycling [21], while the non-labile (<53 μm) or recalcitrant pool which has a long turnover is responsible of carbon sequestration and climate regulation [22]. Soil fertility in terms of nutrient availability is sensitive to the labile pools of SOC, whereas soil potentiality for climate regulation through carbon sequestration and ecosystem sustainability depends much more on the non-labile pool of SOC [23]. Furthermore, discussions by the United Nations Framework Convention on Climate Change (UNFCCC) and other international fora placed agricultural soil in a vital position for mitigating climate change [24,25]. Hence, soil management practices that promote carbon storage are essential for sustainable production systems and climate change adaptation and mitigation. Therefore, understanding the dynamics of soil organic carbon stock and its subsequent pools in different cropping systems is essential for designing and implementing sustainable farming systems [26].

Cropping system specifications and farm management practices have great impact on carbon storage and its spatiotemporal kinetics [27,28]. Soil potential for organic matter storage depends on soil type, soil management practices and climate conditions [2,29,30]. Precipitation positively affects SOC content, while temperature adversely affects SOC vertical distribution [31]. Previous studies [29,32–38] have reported that farm management practices such as crop residues restitution, mulching, organic amendment, cover crop, legume intercropping and biochar application to soil have positive effects on soil organic carbon storage. However, the quality of the organic resources used is a key parameter of carbon storage in the soil. Choudhury et al. [39] and Yoo and Wander [40] demonstrated that tillage leads to soil aggregate break-up and soil organic carbon mineralization, while no tillage induces higher soil particle aggregation, carbon sequestration and particulate organic matter (POM) buildup [41].

In Benin, studies evaluating the effect of cropping systems on SOC are limited to a few studies reporting on agroforestry systems [42,43], cereal–legume-based cropping systems [14,44], palm oil-based cropping systems [33,45,46], vegetable cropping systems under poultry and sheep dung manures and fallow land [32,42]. Despites that, the dynamics of soil organic carbon stock (C stock) and its pools are at the center of various discussions on climate and sustainable development, with more research interest over the last decades in Benin, very few studies have reported on smallholder farming systems which are very complex in terms of resource endowment and integration with various spatiotemporal

arrangements [47,48]. Knowledge related to the effects of different farming systems on the C stock in the region is still unclear and limited to a few research studies [32,33,42,43,45,46,49]. Therefore, the objectives of this study were to (i) investigate the C stock across selected cropping systems and (ii) assess the particle-size distribution of SOC in these cropping systems at a watershed scale. We hypothesized that fallow land and teak plantation store more carbon than cultivated lands and the POM fraction is more sensitive to cropping system characteristics than the NOM.

## 2. Materials and Methods

### 2.1. Study Area

The study was carried out in Kiti sub-watershed in central Benin. The sub-watershed is part of the Zou watershed which is one of the biggest watersheds of Benin. The sub-watershed of Kiti lies between $2°4'00''$–$2°12'00''$ longitude East and $7°20'00''$–$7°29'00''$ latitude North and covers an area of 85,690.8 ha (Figure 1). The mainstream of the sub-watershed is Kiti which is a tributary of the Zou River. The climate in this area is tropical Sudano-Guinean, with a bimodal rainfall pattern. Daily temperatures range from 26 to 31 °C, and annual rainfall averages range between 1000 to 1200 mm [43]. Soils are primarily ferruginous tropical soil with concretions [50] classified as Luvisols [51]. These soils are characterized by a yellowish to light brown sandy horizon on brownish red clay, very concretionary with angular quartz gravels and occasionally ferruginous [52]. They have substantial alterations with an accumulation of ferric hydrates associated with very little oxidized aluminum [52]. The texture is sandy-clay with poor drainage at deeper layers due to high clay eluviation from the surface layer [11]. The sub-watershed of Kiti is part of the Central Benin cotton agroecological zone (ZAE 5). The vegetation is a lightly wooded savannah with sparse shrubs of natural trees and small-sized plantations, with agriculture being the predominant livelihood means for the communities around the watershed. This sub-watershed was purposely chosen because it is an area of intensive agricultural production of cash crop (e.g., cotton) and staple food crops (e.g., maize) in the Zou watershed with a substantial impact on smallholder farmers' livelihoods.

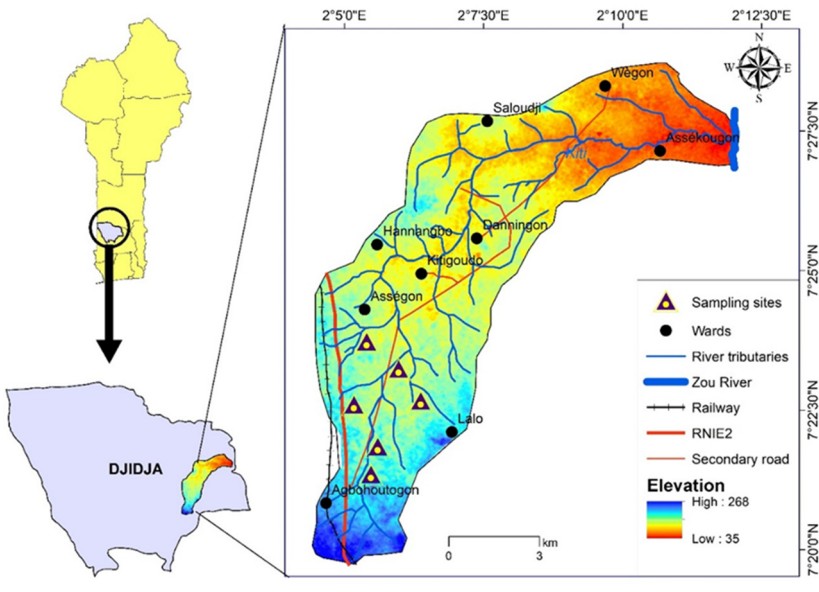

Source: Topographic map IGN (1992); Fieldwork (2017)

**Figure 1.** Map showing the sampling sites.

### 2.2. Cropping Systems

As the primary potential characteristic of this agroecological zone, cotton (*Gossypium* spp.) is the main cash crop cultivated in this region. The cropping systems are dominated by maize (*Zea mays* L.) and cotton-based cropping systems. Maize–cotton relay cropping (MCRC) is the primary cropping system implemented in the watershed. The maize–cotton

system is a relay cropping system characterized by manual ploughing at a maximum depth of 20 cm. Maize is sown at the beginning of the long rainy season (between mid-March and mid-April), while cotton is sown in the maize cob maturity stages (between 15 July and 30 August). The use of mineral fertilizer is globally low for maize and other crops, while 150 kg/ha of NPK (15-15-15) and 50 kg/ha of urea (46% N) is usually applied for cotton. Yam (*Dioscorea* spp.) is grown in the watershed in small plots. As yam cropping requires high soil fertility, it is generally cultivated at the top of the crop rotation on new fallow land. Thus, yam producers constantly look for new fallow or forest lands to convert into farmland [53]. The yam–maize intercropping system (YMI) is characterized by mound ploughing at about 40 cm high. In the sub-watershed of Kiti, yam is intercropped with maize with low or no fertilizer input. In the cropping systems MCRC and YMI, maize and yam residues are spread in the furrows while cotton stalks are gathered and burned for pest management. Although, most of the farmers in the watershed leave the crop residues on the farm as mulch for replenishing soil fertility, these crop residues are commonly grazed by livestock belonging to transhumant pastoralists passing through the region in search of graze for their livestock, especially during the dry season [54–56]. This leads to almost a complete exportation of crop residue from the farm, leaving bare the soil surface, which becomes more susceptible to erosion from heavy winds during the dry season and rain at the beginning of the wet period [57]. The lands which have higher proportion of soil concretion are difficult to plough and are generally used for tree plantation, notably teak plantation (*Tectona grandis*). The teak plantation investigated in this study is a plantation established since 1998. The wooded trees are sold for use as posts or poles with a diameter of 5 to 15 cm and an average harvesting period ranging from 5 to 10 years [58]. Although the fallow period has generally reduced due to land shortage, some farmers in the area still observe a fallow period ranging from 5 to 10 years and above. During farmland exploration, two typical fallow lands have been identified: 5-year fallow (5YF) and above 10-year fallow (Ab10YF). The fallow lands, 5YF and Ab10YF, were covered by natural vegetation and shrubs, including *Vitellaria paradoxa*, *Azadirachta indica*, *Nauclea latifolia*, *Danielia oliveri*, *Imperata cylindrica* and *Cleome viscosa*. However, these fallow lands are influence by seasonal vegetation fires during the dry season [59,60].

### 2.3. Sample Collection and Analysis

The study used an experimental research design considering the cropping systems as the principal factor to investigate. The cropping systems included maize–cotton relay cropping (MCRC), yam–maize intercropping (YMI), teak plantation (TP), five-year fallow (5YF) and above ten-year fallow (10YF) (Table 1). Soil samples were collected from July to August 2017, using a grid establishment approach [61]. The grids were constructed using a step of 20 m × 30 m on a total area of 6 sq.km covering the five cropping systems investigated. The grids for soil sample collection were selected randomly in each cropping system. A total of 50 grids were sampled: 18 grids in MCRC, 8 in YMI, 8 for TP, 8 in 5YF and 8 in Ab10YF. The high number of sampling grids in MCRC compared to the others was because of the high coverage of this cropping system across the sub-watershed. Soil samples were collected at four depths in each grid: 0–10, 10–20, 20–30 and 30–40 cm. The collected soil samples were air-dried for three weeks, mechanically crushed using a stainless-steel roller and sieved through a 2 mm sieve for laboratory analyses. Soil organic carbon content in soil was determined using boiled potation bichromate in acidic conditions, as described in Okalebo et al. [62]. The absorbance of the samples was read with a spectrophotometer at a wavelength of 600 nm. Furthermore, the soil pH was determined in a distilled water ratio of 1:2.5 and the Robinson pipette method was used for soil texture determination [63]. In each sampled grid, the cylinder method (calibrated density cylinders of a known volume of 100 cm$^3$) was used to collect the samples for soil bulk-density determination for each layer. The contents of the cylinder were weighed after drying at 105 °C in an oven for 24 h. The bulk density BD is given by the ratio of dry weight to volume.

**Table 1.** Cropping systems history.

| Years | Cropping Systems | | | | |
|-------|------|-----|-----|-----|--------|
|  | **MCRC** | **YMI** | **TP** | **5YF** | **Ab10YF** |
| 2005–2006 | Fallow | Maize–cotton | Teak plantation | Maize–cotton | Fallow |
| 2006–2007 | Fallow | Maize–cotton | Teak plantation | Maize–cotton | Fallow |
| 2007–2008 | Maize–soybean | Maize–soybean | Teak plantation | Maize–cotton | Fallow |
| 2008–2009 | Maize–soybean | Maize–soybean | Teak plantation | Maize–cotton | Fallow |
| 2009–2010 | Maize–soybean | Fallow | Teak plantation | Maize–cotton | Fallow |
| 2010–2011 | Maize–cotton | Fallow | Teak plantation | Maize–cotton | Fallow |
| 2011–2012 | Maize–cotton | Fallow | Teak plantation | Maize–cotton | Fallow |
| 2012–2013 | Maize–cotton | Fallow | Teak plantation | Fallow | Fallow |
| 2013–2014 | Maize–cotton | Fallow | Teak plantation | Fallow | Fallow |
| 2014–2015 | Maize–cotton | Fallow | Teak plantation | Fallow | Fallow |
| 2015–2016 | Maize–cotton | Yam–maize | Teak plantation | Fallow | Fallow |
| 2016–2017 | Maize–cotton | Yam–maize | Teak plantation | Fallow | Fallow |

*2.4. Soil Carbon Stock Calculation*

Soil organic carbon stock computation was based on soil bulk-density, the thickness of the soil layer and the proportion of fine soil. The C stock was computed using Equation (1) [42]. Carbon stock was estimated for the four soil layers 0–10, 10–20, 20–30 and 30–40 cm. C stock was considered for each layer, for 0–30 cm and for 0–40 cm.

$$\text{C stock} = \sum_{Depth=1}^{Depth=n} \text{C stock}_{Depth} = \sum_{Depth=1}^{Depth=n} (\text{SOC} \times \text{BD} \times P \times (1 - \text{frag}) \times 10) \tag{1}$$

where C stock (Mg C·ha$^{-1}$) is the sum of soil organic carbon stock for the different layers considered, C stock$_{Depth}$ (Mg C·ha$^{-1}$) is the stock of organic carbon at a specific soil depth, SOC (g·C kg$^{-1}$) is the concentration of soil total organic carbon, BD (g·cm$^{-3}$) is soil bulk-density, P (m) is the thickness of the soil layer, frag is the percentage volume of coarse fragments/100 and n is the number of layer considered.

*2.5. Soil Organic Matter Particle-Size Fractionation*

There are various methods for soil organic carbon partitioning into its granulometric functional pools. The particle-size fractionation of soil organic carbon allows to assess the distribution of SOC according to particle sizes. It gives useful information on the proportion of the different types of soil organic matter, their chemical composition, and potential dynamics in soil which are essential in evaluating the sustainability of various land management options for soil organic carbon rehabilitation. Soil organic carbon fractionation used in this study was adapted from the method develop by Feller [64] for coarse texture and poor humus content soil, used with good overall accuracy by Sainepo et al. [65] in Kenya; by Koussihouèdé et al. [32] in Benin and Gura et al. [66] in South Africa.

A reciprocal shaker was used to mix 50 g of soil with 300 mL of distilled water with 10 mL of Calgon solution (10% sodium hexametaphosphate, 50 g·L$^{-1}$) for 15 h. The solution was passed through a series of nested sieves of sizes 2000 μm, 250 μm and 53 μm in a wet sieving apparatus with deionized water. The particles that passed through the 53 μm was referred as the non-particulate organic matter fraction (NOM). The fraction 53–250 μm was referred as the fine particulate organic matter fraction (fPOM), while the 250–2000 μm fraction was considered as the coarse particulate organic fraction (cPOM) [67]. In the cPOM fraction, plant materials such as plant residues and roots that had partially broken down were carefully separated. The isolated particles for cPOM and fPOM were washed with deionized water until clean and backwashed into an evaporation dish. The fraction that passed through the 53 μm sieve was collected in a volumetric flask, quantified, thoroughly homogenized and a sample of 100 mL was collected in an evaporation dish. The evaporation dishes were dried at 65 °C till a constant weight. The oven-dried soil particles were weighed and placed in dry porcelain crucibles and heated in a muffle furnace

at 450 °C for 4 h to separate the mineral particles from the organic particles. After cooling, the organic matter contained in each fraction was determined as shown in Equation (2).

$$\text{Fraction} = \frac{\text{Weight at 65 °C} - \text{Weight at 450 °C}}{\text{Weight at 65 °C}} \tag{2}$$

The organic carbon content in the fractions were calculated using the coefficient 1.724 considering that organic matter comprises 58% carbon [68]. Particle-size fractionation of SOC was caried out for two soil layers, 0–10 and 10–20 cm. For each sample the percentage recovery was calculated by the ratio of the sum of the weight of the three fractions by the initial weight of 50 g, multiplied by 100 (Equation (3)) and the total fraction was reported considering 1000 g of soil. The enrichment factor (EF) in each of the fractions was calculated according to Equation (4) [69].

$$\% \text{ Recovery} = \frac{\text{Weigth cPOM} + \text{WeigthfPOM} + \text{WeigthNOM}}{50} \times 100 \tag{3}$$

$$\text{EF} = \frac{\text{SOC fraction (g/kg)}}{\text{totalSOC (g/kg)}} \times 100 \tag{4}$$

*2.6. Statistical Analysis*

The dataset was screened for normality and variance homogeneity using the Shapiro–Wilk test [70] and Bartlett's test [71]. Thereafter, a one-way ANOVA was used to determine significant differences among the different cropping systems. Tukey's post hoc test was used to separate the means, in case of significant difference at a 5% significance threshold level. All analyses were performed in R software version 4.1.

**3. Results**

*3.1. Soil Properties*

In this study, there was no significant difference in soil texture (p = 0.09) across the cropping systems and soil was sandy-clay loam or sandy-clay depending on their silt content (Table 2). Soil pH varied significantly (*p* = 0.03) between the cropping systems and soil under MCRC exhibited the lowest pH values. The soil under MCRC was classified as acidic while the soil under other cropping systems were classified as slightly acidic. Soil bulk-density (BD) values were significantly higher for the surface depth layers of 0–10 and 10–20 cm in the soil under fallow (5YF and Ab10YF) and teak plantation than the soil under cropping (MCRC and YMI). However, for the sub-layer depths (20–30 and 30–40), regardless of the cropping system, BD values increased with YMI showing the lowest BD while MCRC, TP, 5YF and Ab10YF exhibited the highest values.

**Table 2.** Soil properties under the different cropping systems.

| Soil Properties | Soil Depth | MCRC | Yam-Maize | TP | 5YF | Ab10YF | *p*-Value |
|---|---|---|---|---|---|---|---|
| **Clay (g·kg$^{-1}$)** | 0–20 | 333.13 ± 38 a | 389.67 ± 52 a | 344.13 ± 45 a | 366.36 ± 48 a | 379.94 ± 85 a | 0.090 [ns] |
| **Silt (g·kg$^{-1}$)** | 0–20 | 31.90 ± 36 a | 27.20 ± 44 a | 28.90 ± 30 a | 39.59 ± 66 a | 39.67 ± 52 a | 0.310 [ns] |
| **Sand (g·kg$^{-1}$)** | 0–20 | 636.80 ± 69 a | 599.77 ± 63 a | 624.80 ± 57 a | 595.32 ± 75 a | 583.47 ± 86 a | 0.100 [ns] |
| **Soil texture** | 0–20 | Sandy clay loam | Sandy clay | Sandy clay loam | Sandy clay | Sandy clay | - |
| **pH** | 0–20 | 5.80 ± 0.2 b | 6.05 ± 0.3 ab | 6.1 ± 0.1 ab | 6.2 ± 0.2 a | 6.3 ± 0.1 a | 0.010 * |
| **BD** | 0–10 | 1.27 ± 0.17 ab | 1.41 ± 0.27 a | 1.14 ± 0.08 b | 1.43 ± 0.21 a | 1.11 ± 0.18 b | 0.024 * |
| | 10–20 | 1.46 ± 0.16 b | 1.55 ± 0.22 b | 1.61 ± 0.09 ab | 1.54 ± 0.08 ab | 1.68 ± 0.01 a | 0.009 * |
| | 20–30 | 1.67 ± 0.12 a | 1.47 ± 1.14 b | 1.71 ± 015 a | 167 ± 0.21 a | 1.69 ± 0.013 a | 0.032 * |
| | 30–40 | 1.73 ± 0.2 a | 1.61 ± 0.2 b | 1.67 ± 0.13 a | 1.68 ± 0.18 a | 1.77 ± 0.08 a | 0.040 * |

MCRC: maize–cotton relay cropping; 5YF: 5-year fallow; Ab10YF: above 10-year fallow; YMI: yam–maize intercropping system; TP: teak plantation; BD: bulk density. Means that do not share a letter are significantly different at $\alpha$ = 0.05. [ns] non-significant at 5%; * *p* value significant at 5%.

### 3.2. Total Soil Organic Carbon Content (SOC) and Soil Organic Carbon Stock (C Stock) across the Cropping Systems

The variations in SOC and C stock from the four layers, 0–10, 10–20, 20–30 and 30–40 cm are presented in Table 3. For the soil layer 0–10 cm, the SOC content significantly varied ($p < 0.001$) between the cropping systems and ranged from 3.14–24.1 g C·kg$^{-1}$. The highest SOC was recorded with Ab10YF while the lowest was recorded with MCRC. In the layer 10–20 cm, the SOC content showed significant ($p = 0.011$) differences between the cropping systems with 5YF and YMI recording the highest and the lowest SOC, respectively (5.22 and 2.21 g C·kg$^{-1}$). At the soil layer of 20–30 cm, the SOC content between the cropping systems was not significantly different ($p = 0.1$) with values ranging between 2.10–2.84 mg C·kg$^{-1}$. In the soil layer of 30–40 cm, the SOC showed significant variation ($p = 0.021$) with values ranging from 1.36–4.43 g C·kg$^{-1}$. The cropping systems YMI and 5YF exhibited the lowest and the highest concentrations of SOC, 1.36 and 4.43 g C·kg$^{-1}$, respectively.

**Table 3.** Soil organic carbon content and carbon stock per soil depth layer.

| Soil Properties | Depth (cm) | Cropping Systems | | | | | *p*-Value |
| --- | --- | --- | --- | --- | --- | --- | --- |
| | | MCRC | YMI | TP | 5YF | Ab10YF | |
| **SOC** **(g·kg$^{-1}$)** | 0–10 | 3.14 ± 0.98 b | 7.37 ± 4.24 b | 5.64 ± 2.62 b | 4.94 ± 2.3 b | 24.1 ± 11.6 a | <0.001 *** |
| | 10–20 | 3.03 ± 1.33 b | 2.21 ± 0.47 b | 3.16 ± 1.04 ab | 5.22 ± 3.78 a | 2.43 ± 0.23 b | 0.011 * |
| | 20–30 | 2.55 ± 0.79 a | 2.06 ± 0.55 a | 2.84 ± 1.0 a | 2.83 ± 1.04 a | 2.10 ± 0.43 a | 0.1 ns |
| | 30–40 | 2.24 ± 0.61 ab | 1.36 ± 0.29 b | 2.64 ± 0.85 ab | 4.43 ± 4.37 a | 2.27 ± 2.47 ab | 0.021 * |
| **C stock** **(Mg C·ha$^{-1}$)** | 0–10 | 3.25 ± 0.52 c | 7.34 ± 3.64 bc | 4.72 ± 1.9 bc | 5.73 ± 2.35 bc | 18.1 ± 6.35 a | <0.001 *** |
| | 10–20 | 3.0 ± 0.7 ab | 2.98 ± 0.63 ab | 3.31 ± 2.05 ab | 6.29 ± 5.0 a | 1.39 ± 0.5 b | 0.001 ** |
| | 20–30 | 2.90 ± 1.07 a | 2.90 ± 1.07 a | 3.08 ± 1.13 a | 2.47 ± 0.91 ab | 1.35 ± 0.1 b | 0.006 ** |
| | 30–40 | 2.24 ± 0.49 ab | 1.81 ± 0.57 ab | 1.97 ± 0.96 ab | 3.26 ± 2.26 a | 1.35 ± 0.57 b | 0.012 * |

MCRC: maize–cotton relay cropping; 5YF: 5-year fallow; Ab10YF: above 10-year fallow; YMI: yam–maize intercropping system; TP: teak plantation; SOC: soil organic carbon; C stock: carbon organic stock. Means that do not share a letter are significantly different at $\alpha = 0.05$; * *p* value significant at 5%; ** *p* value significant at 1%; *** *p* value significant at 0.1%. ns: non-significant at 5%

The C stock showed significant variation between the cropping systems and soil depth layers ($p < 0.001$, $p = 0.001$, $p = 0.006$, $p = 0.012$). Regardless of the cropping systems, C stocks were higher for the surface layer 0–10 cm and showed a decreasing trend towards sub-surface layers. For the soil layer 0–10 cm, Ab10YF recorded the highest C stock (18.1 Mg C·ha$^{-1}$), while MCRC recorded the lowest (3.25 Mg C·ha$^{-1}$). In the layer of 10–20 cm, 5YF showed the highest C stock (6.29 Mg C·ha$^{-1}$), while Ab10YF recorded the lowest C stock 1.39 Mg C·ha$^{-1}$ and the cropping systems ranged in the order 5YF > TP > MCRC > YMI > Ab10YF. In the layer of 20–30 cm, C stock in the soil ranged from 1.35 to 3.26 Mg C·ha$^{-1}$ and Ab10YF and 5YF exhibited the lowest and the highest C stocks, respectively. In the layer of 30–40 cm, the C stock ranged between 3.26 and 1.35 Mg C·ha$^{-1}$ with Ab10YF and 5YF recording the lowest C stock values. In addition, there was a significant ($p = 0.012$ and $p = 0.02$) difference between the cropping systems for the C stock at the layer of 0–30 cm and at the layer 0–40 cm (Figure 2). Considering the total C stock of the layer 0–40 cm, the cropping systems Ab10YF exhibited the highest C stock (22.20 Mg C·ha$^{-1}$) while MCRC recorded the lowest (10.31 Mg C·ha$^{-1}$). Considering the total C stock for the surface layer of 0–30 cm (Figure 2), the highest C stock was recorded at Ab10YF and the lowest at MCRC 20.84 and 9.23 Mg C·ha$^{-1}$, respectively.

### 3.3. Organic Carbon Concentrations in Particle Size Fractions

The average total fraction masses ranged between 969.33 and 991.84 mg·frac·g$^{-1}$ soil indicating an average recovery rate varying between 96.9 and 99.1% (Table 4). Regardless of the cropping system and the soil layer, the non-particulate organic fractions (NOM) showed the highest carbon concentration 3.40–7.99 mg·kg$^{-1}$. The fine particulate organic fractions (fPOM) and the coarse particulate fractions (cPOM) depicted the lowest carbon content between 0.56 and 2.3 mg/kg. In the layer 0–10 cm, carbon concentration in the cropping systems Ab10YF and 5YF were significantly higher in the three fractions NOM ($p = 0.003$),

fPOM ($p = 0.01$) and cPOM ($p = 0.02$). The cropping systems YMI and MCRC recorded the lowest carbon concentrations. For the layer 10–20 cm, 5YF exhibited the highest carbon content in NOM, while MCRC and YMI showed the lowest (1.05 and 1.26 mg·kg$^{-1}$, respectively). For this layer, SOC concentration in the NOM fraction varied significantly between the cropping systems ($p = 0.04$). No significant difference was observed for the carbon concentration in the fPOM fraction. However, the carbon content in the cPOM fraction varied significantly ($p = 0.04$) between the cropping systems, with Ab10YF and 5YF and TP recording the highest carbon concentrations 1.24, 1.12 and 0.84 g/kg, respectively.

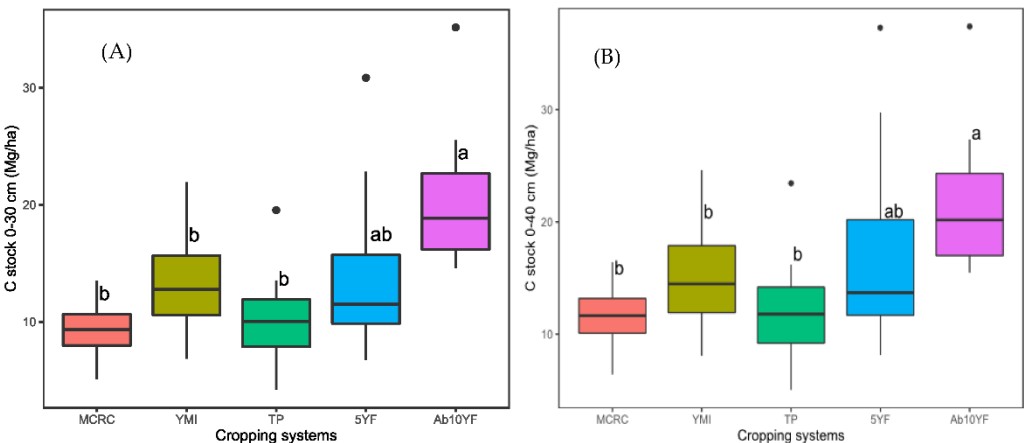

**Figure 2.** Total carbon stock for soil layers 0–30 cm (**A**) and 0–40 cm (**B**) MCRC: maize–cotton relay cropping; 5YF: 5-year fallow; Ab10YF: above 10-year fallow; YMI: yam–maize intercropping system; TP: teak plantation. Means that do not share a letter are significantly different at α = 0.05.

**Table 4.** Carbon concentrations in particle size fractions across the cropping systems for 0–10 and 10–20 cm.

| Depth (cm) | Fractions | MCRC (g·kg$^{-1}$) | YMI (g·kg$^{-1}$) | TP (g·kg$^{-1}$) | 5YF (g·kg$^{-1}$) | Ab10YF (g·kg$^{-1}$) | *p* Value |
|---|---|---|---|---|---|---|---|
| **0–10** | NOM | 3.40 ± 0.4 d | 4.46 ± 0.38 c | 3.73 ± 0.33 d | 5.93 ± 0.39 b | 7.99 ± 0.21 a | 0.003 ** |
| | fPOM | 1.19 ± 0.21 b | 1.23 ± 0.32 b | 1.28 ± 0.3 b | 1.89 ± 0.2 ab | 2.24 ± 0.12 a | 0.01 * |
| | cPOM | 0.70 ± 0.4 b | 0.71 ± 0.42 b | 1.08 ± 0.9 ab | 2.09 ± 0.8 a | 2.3 ± 5.33 a | 0.02 * |
| **Total fraction mass (g·kg$^{-1}$ soil)** | | 986.62 | 978.67 | 971.22 | 981.45 | 969.33 | |
| **10–20** | NOM | 1.05 ± 0.38 c | 1.26 ± 0.25 c | 1.66 ± 0.30 b | 2.53 ± 0.27 a | 1.81 ± 0.99 b | 0.04 * |
| | fPOM | 1.14 ± 0.47 a | 0.99 ± 0.29 a | 0.78 ± 0.21 a | 0.92 ± 0.29 a | 0.98 ± 0.33 b | 0.044 * |
| | cPOM | 0.56 ± 0.11 b | 0.41 ± 0.21 b | 0.84 ± 0.32 ab | 1.12 ± 0.17 a | 1.24 ± 0.13 a | 0.04 * |
| **Total fraction mass (g/kg soil)** | | 991.84 | 988.33 | 975.52 | 978.56 | 989.42 | |

MCRC: maize–cotton relay cropping; 5YF: 5-year fallow; Ab10YF: above 10-year fallow; YMI: yam–maize intercropping system; TP: teak plantation; fPOM: fine particulate organic matter; cPOM: coarse particulate organic fraction; NOM: non-particulate organic matter. Means that do not share a letter are significantly different at α = 0.05; * *p* value significant at 5%; ** *p* value significant at 1%.

### 3.4. Carbon Enrichment Factor (EF) in Particle-Size Fractionation

The contribution of each particle-size organic matter fraction to the total organic carbon content expressed using the enrichment factor for the two layers revealed that regardless of the cropping systems and the layer, the NOM fraction exhibited the greatest contribution to the total SOC, while cPOM exhibited the lowest contribution (Figure 3). In the two layers, MCRC recorded the highest contribution of the NOM fraction 71.9% and 75.03%. The cropping systems Ab10YF, YMI and TP recorded the highest carbon contribution from the cPOM between 7.1% and 22% for the two layers and between 28.89% and 44.41% for fPOM. In addition, EF values showed that the cPOM and fPOM were the most influenced by the cropping systems.

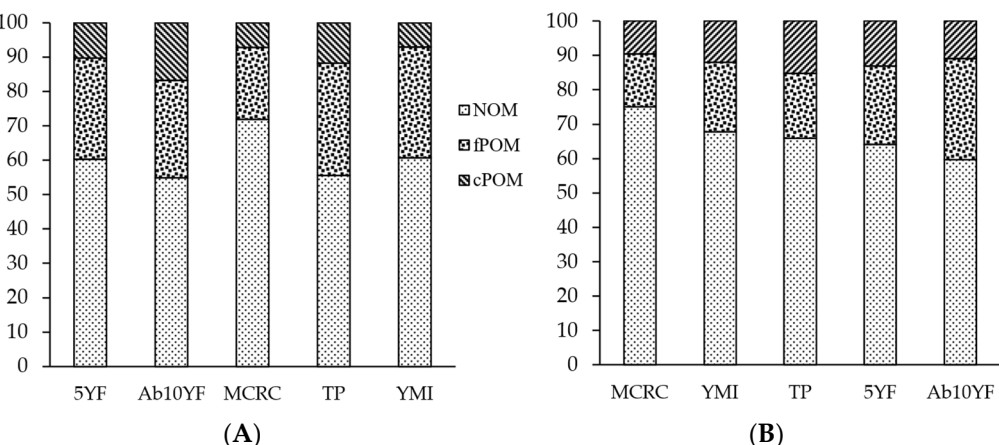

**Figure 3.** Carbon enrichment factor EF of the organic matter fractions (**A**) 0–10 cm layer, (**B**) 10–20 cm layer. MCRC: maize–cotton relay cropping; 5YF: 5-year fallow; Ab10YF: above 10-year fallow; YMI: yam–maize intercropping system; TP: teak plantation.

## 4. Discussion

The present study assessed the soil organic carbon stock and its distribution in three particle-size fractions considering five cropping systems in the Kiti sub-watershed in the Zou watershed in central Benin. This paper contributes to a growing understanding of the dynamics of soil organic carbon storage in coarse structure tropical soils in sub-Saharan Africa (SSA). The C stock in this study was estimated using the method considering the sum of the stocks of the different layers of the soil profile (0–10, 10–20, 20–30 and 30–40 cm), known as the classical method for C stock calculation. The limitation of this method is that it does not consider the variations in soil mass. To curb this, Ellert et al. [72] and Arrouays et al. [73] introduced soil equivalent masses and suggested C stocks estimation by the equivalent masses rather than the estimation by soil depth. This approach allows to reliably assess the changes in organic matter quantities linked to time or soil management practices. This method of calculation was used by Barthès et al. [14]; Aholoukpè [33] and Houssoukpèvi et al. [45].

The C stock recorded in this study ranged between 9.23 and 20.84 Mg C·ha$^{-1}$ for the layer 0–30 cm and between 11.48 and 22.20 Mg C·ha$^{-1}$ for the layer 0–40 cm. The carbon stock recorded in this study was slightly smaller than those recorded by previous studies in Benin [32,33,38,45,74]. However, the stocks recorded were higher than those recorded by Saidou et al. [42]. The low stock recorded in this study compared to previous studies could be attributed to various factors including tillage, the absence of crop residue restitution and seasonal vegetation fires which have been proven to negatively influence soil organic carbon stocks [74]. Moreover, the low stock observed in the study could also be attributed to the high proportion of concretion in the soil which can lead to a low proportion of fine particles and consequently affect soil organic carbon stocks. This is in line with research by Hairiah et al. [75] and Reichenbach et al. [76] who illustrated that the geochemical properties of the soil parent material leave a footprint that affects SOC stocks and mineral-related C stabilization mechanisms.

The soil under fallow (5YF and Ab10YF) had the highest C stock compared to teak plantation (TP) and the croplands (MCRC and YMI). The results are consistent with other work [32,77,78], highlighting that soils under fallow are enriched in organic matter from decaying litter, leaves and branches with lignified materials, which decompose progressively and replenish the soil organic carbon pool. The C stocks recorded in this study were low compared to those recorded by [75] (13.68; 12. 73 and 24.40 Mg C. ha$^{-1}$ for the layer 0–20 cm) on vegetable farmland receiving organic amendment (poultry manure and sheep dung) and a 5-year fallow. These differences could be attributed to differences in farm management practices implemented in regard to vegetable farming versus staple and cash crop farming. Previous studies assessing the effect of farm management practices revealed

that farm management practices, including a fallow period from five years and above, soil amendments, cover crops, mulching and crop residues restitution have a positive effect on C stocks in the soil [2,29,32,33]. The cultivated lands (MCRC and YMI) recorded the lowest C stocks. This could be attributed to the low crop residue restitution and the tillage system which can induce soil aggregate crumbling and therefore rapid carbon mineralization. The C stock observed in the soil under teak plantation was lower compared to the 30.5 and 31.4 Mg C·ha$^{-1}$ recorded by Houssoukpèvi et al. [45] for cropland and tree plantations in southern Benin. This could be attributed to the age of the plantation and the uneven exploitation scheme, making the plantation have a scattered structure and therefore a lower carbon input. Since existing studies reporting on C stocks in Benin were conducted at different layers making their comparison challenging, an extrapolation offers the possibility to compared different cropping systems on the basis of the surface layer of 0–30 cm. This extrapolation depicted that for the Acrisol in southern Benin, C stocks of 73 Mg C·ha$^{-1}$ under fallow, 41 Mg C·ha$^{-1}$ under vegetable farming systems with chicken manure, 38 Mg C·ha$^{-1}$ with sheep ruminant dung [32] and 32 Mg C·ha$^{-1}$ under the maize–mucuna cropping system [14]. This confirms that farm management practices have a substantial effect on soil organic carbon stocks. Since limited studies have focused on C stock evaluation across different farming systems, different agroecological zones, and different soil types as well as the long-term influence of these systems, more in-depth studies will help to identify and implement sustainable farming systems for better carbon sequestration and resilient food systems. In addition, as watersheds have a proven propensity to soil erosion [79], understanding the impact of landform on soil organic carbon storage could have a great contribution to developing sustainable farming systems at the watershed scale.

　　　The particle-size distribution of organic carbon in the different particle-size fractions indicated that the non-particulate organic carbon fraction, associated with silt-clay (<53 μm), held the largest contribution to the total organic carbon. The contributions of the coarse particulate organic matter (cPOM) to the total soil organic carbon reserves were the lowest in all the cropping systems. The carbon associated with the organo-mineral fraction are localized in clay and silt bonds which protect the carbon from mineralization. Indeed, under conditions highly favorable to biological decomposition and humification, such as those in tropical regions, particulate organic matter is exposed to mineralization processes and therefore represents a small portion of the total organic carbon pool in the soil [80]. The two particulate fractions, cPOM and fPOM, have been proven to be the most affected by cropping systems [81]. These results are consistent with previous studies that emphasized the vulnerability of this fraction to mineralization processes [65,81]. These fractions, being free from soil mineral particles, are more accessible to microorganisms. The low cPOM and fPOM in the cultivated land, could be explained by tillage and soil erosion. Tillage induces soil aggregate breakdown and accelerates organic carbon mineralization. Furthermore, the cPOM and fPOM are lighter fraction and therefore susceptible to be streamed away during storm rain. For example, Akplo [79] pointed out in the Zou watershed that the particulate portion (cPOM + fPOM) of soil organic carbon was significantly affected by soil erosion.

　　　Although the carbon content in the biomass and soil amendments applied to soil are generally accumulated more in the fPOM and cPOM fractions, the long-term accumulation of carbon in soil is predominantly determined by the carbon in the silt and clay fractions, NOM [82]. The higher concentration of the cPOM and fPOM in the two fallow lands can be attributed to the shoots and residues from the thick vegetation that accumulated during the fallow period. The fine compartment organic matter (NOM) is associated with micro-aggregates that protect organic carbon by adsorption and occlusion on mineral surfaces. The abundance of fine elements favors stabilization of soil organic carbon at a higher level of dynamic equilibrium, good structural stability of soil and makes the system more sustainable. According to our results, NOM is significantly higher in the fallow land. The biologically and chemically active fractions, fPOM and cPOM, belonging to the labile compartment is very sensitive to cultivation practices [83]. Ploughing is considered

as an unfavorable factor for the storage of organic matter in the soil [84]. It favors the destruction of soil micro-aggregates and accelerates soil organic carbon mineralization. This explains the low concentrations observed in the cultivated lands. Long-term fallows remain good sustainable land management practices. However, in the current context, where croplands are shrinking in favor of urbanization and development, yet food need is growing and cropland expansion is limited, there is a need for in-depth studies establishing more sustainable and resilient intensification of farming systems [85,86].

## 5. Conclusions

In this paper, we assessed soil organic carbon stocks and its particle-size fractionation across different cropping systems. The fallow lands, Ab10YF and 5YF, exhibited the highest C stocks 17.74 and 22.20 mg C·ha$^{-1}$, while cultivated land under tillage MCRC depicted the lowest C stocks. The three organic carbon fractions showed significant variation across the cropping systems, with the NOM fraction holding the largest contributions to total soil organic carbon for all the cropping systems. The cPOM and fPOM were most influenced by the cropping systems with the highest concentration observed in Ab10YF and 5YF. The hypotheses established at the beginning of the study, stating that fallow land and teak plantations store more carbon than cultivated lands and that the POM fraction is more sensitive to cropping system characteristics than NOM is partially confirmed since the teak plantation recorded a lower C stock. The implementation of sustainable soil fertility management practices, including efficient restitution of crop residues, mulching, legume intercropping, long rotation cycles, use of dual-purpose crops and avoiding seasonal fires are necessary to improve soil organic storage in agricultural soil in the watershed. Moreover, agricultural extension officers, policy-makers and officials of the ministry in charge of agriculture have the significant role in supporting farmers in the sustainable intensification of cropping systems and the regulation and enforcement for the respect of pastoral transhumance corridors.

**Author Contributions:** Conceptualization: A.M.A.A., F.K.A. and T.M.A.; data curation: A.M.A.A. and T.M.A.; formal analysis: A.M.A.A. and F.K.A.; finding acquisition and investigation: F.K.A. and T.M.A.; methodology: A.M.A.A., F.K.A. and T.M.A.; project administration: F.K.A., H.A.A. and P.H.; supervision: F.K.A.; validation: H.A.A. and P.H.; visualization: F.K.A., T.M.A.; writing—original draft: A.M.A.A. and T.M.A.; writing—review and editing: A.M.A.A., F.K.A., T.M.A., H.A.A. and P.H. All authors have read and agreed to the published version of the manuscript.

**Funding:** This research received no external funding.

**Institutional Review Board Statement:** Not applicable.

**Informed Consent Statement:** Not applicable.

**Data Availability Statement:** Data available upon request from the corresponding author.

**Acknowledgments:** We are grateful to the International Atomic Energy Agency (IAEA), which provided the equipment used for laboratory analysis, through the Regional Project RAF5075.

**Conflicts of Interest:** The authors declare that they have no known competing financial interests or personal relationships that could appear to influence the work reported in this paper. The funders had no role in the design of the study; in the collection, analysis, or interpretation of the data; in the writing of the manuscript, or in the decision to publish the results.

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
