# Peer review of "Assessing Soil Organic Carbon Stocks and Particle-Size Fractions across Cropping Systems in the Kiti Sub-Watershed in Central Benin"

_carbon_

Round 1
Reviewer 1 Report
Soil organic carbon storage in agricultural soil constitute a crucial potential for sustainable agricultural productivity and climate change mitigation. The paper aims at assessing soil organic carbon stock and its particle size fractions across five cropping systems in a small watershed. The topic is interesting, however, some aspects should be concerned.
1. The biggest problem of the article is that there may be some mistakes in the data of soil organic carbon and its particle-size fractionation. No matter the content of soil organic carbon or particle-size fractionation, its value is extremely low, which is obviously abnormal. Please check carefully again.
2. There are many format errors in the paper, such as Line 29, the C stock is 22.20 mg C.ha-1? or 22.20 Mg C.ha-1?
Lines 55, One more full stopï¼›Lines 58,There is an extra bracket; Lines 62, Incorrect document citation format. “[13] has shown that.....”. Please carefully check this mistakes throughout the manuscript.
3. There are some conspicuous mistakes in the paper. e.g. Lines 141-143,"The grids were constructed using a step of 20 m x 30 m on a total area of 6 sq.km covering the five cropping systems investigated." 20 m x 30 m=6 km2? Please carefully check this mistakes throughout the manuscript.
4. In the Introduction, the authors need to highlight the novelty of their research.
5. Both the value of total soil organic carbon (section 3.2)and particle-size fractions (section 3.3) is incorrect. Maybe g kg-1 or mg g-1, please check it.
6. Since Table 3 has presented the C stock of each layer of soil, Figure 2 is somewhat repetitive.
Reviewer 2 Report
Reviewer’s comments
This study used five cropping systems located in Kiti sub-watershed in Benin to assess soil organic carbon stock and its distribution in three particle size fractions at a watershed scale. The presentation of the paper is generally good, except a few mistakes in spelling and grammar. But the major concern is what meaningful suggestions the authors bring to the local farmers and policy-makers. The authors should address the concerns in the revised MS.
Abstract
Line 21, In which year were the soil samples collected?
Line 29, tillage MCRC and YMI depicted the lowest C stock. Please give specific values.
Line 31, How much the contribution does the NOM fraction hold?
Line 33, fundings???
Introduction
Lines 44-46, it seems that the agricultural production is not intensified in sub-Saharan Africa (SSA) since local people don’t use modern techniques. Therefore, there exist a contradiction in your expression when you say ‘along with the intensification of agricultural production.’ A transitional sentence is needed to be added between line 45 and line 46.
Line 46, ‘exert pressure’ is not commonly used. Use imposed instead of exerted?
Line 52-53, use of ‘achievement of environmental sustainability and SDG’ is more suitable.
Line 55, drop the additional full stop.
Line 56, you should first tell the readers where Benin is located, then give the readers examples.
Line 57, I can’t figure out the expression ‘poor clay type, kaolinite’, so please revise the sentence.
Line 59, what is miming farming?
Line 60, use constitutes (the third person).
Line 62, use at least two references since you say a number of studies.
Line 62, use Literature [13].
Line 68, use While.
Line 69, use ‘is responsible for’
Line 73, give full name of UNFCCC
There exist several spelling and expression errors in Introduction. The most important part is that, the authors should give a review of the effects of cropping systems on soil carbon storage at home and abroad. However, the authors fail to do this. It is the major drawback of the Introduction section.
2. Materials and methods
Line 97-98, soil texture seems so weird because it simultaneously has the characteristics of sandy soil and clay soil. I am curious about the soil type. Could you present a soil texture table that contains the proportions of each particulate component? Also, you say clay eluviation. The problem is that I can’t figure out whether your phrase means clay has been eluviated from surface layers or the clay soil has eluviation effect.
Line 108, use Italics for Latin name of cotton.
Line 101, use livelihood means, not mean.
Line 116, give the fertilizer name which contains 15-15-15 of NPK, not directly use NPK.
Line 150, drop the parenthesis.
Table 1. What happened on teak plantation after 2007-2008?
Line 180-185, correct L-1
3. Results
Line 219, in the note of Table 2, are you sure α = 0.0 is used for indicating significance?
Line 262, which figure depicts the comparison between predicted and observed rice ETc?
Line 270, confirmed than CNN-SVR is not able in...? maybe there is a mistake in expression.
Line 226-227, you state 5YF and YMI recorded the highest and the lowest SOC respectively, but the values in parenthesis (2.2 and 5.22 mg C.kg-1) are not corresponding the highest and lowest content.
Line 231-232, please give the specific values for the lowest and the highest concentration of SOC.
Line 248, rewrite the title of Table 3 since it is too simple.
Lines 239-242, it is weird C stock was so high in 0-10 cm layer for Ab10YF while it became nearly the lowest values in 10-40 cm layer.
Line 251, what is the unit of C stock in Figure 2? What do the letters mean in Figure 2? What do the open circles and solid black circles mean in Figure 2?
Line 254, the authors didn’t define what the average recovery rate is in Materials and Methods section. I can’t figure out what the relationship is between average total fraction masses and average recovery rate.
Line 266, ... across the Ab10YF, TP and 5YF recording the highest carbon concentration... in this sentence, I can’t figure out which one records the highest carbon concentration.
4. Discussion
Lines 285-299, the whole paragraph is a comparison to the findings of literature [43] and [44], which needs simplifying. The key concern is not addressed by the authors. Why the present results are lower than the previous ones.
Line 309, use staple instead of stapple.
Line 325, ...maize-mucuna cropping system [12] his confirm that farm management practices have a... rewrite the sentence.
Line 326, Grammatical errors in ‘ limited studies has focused’???
Line 332, Grammatical errors in ‘help developing sustainable farming systems’???
5. Conclusions
What is your suggestive advice for local farmers? You say a depth study is needed to establish more sustainable farming system, but what is the meaning of the present study? In conclusion, you should give the local farmers and policy-makers some constructive suggestions from your study.
Author Contributions use abbr of authors’ names.
References
Use correct formats of references.
Reviewer 3 Report
The subject of the article submitted for review is very close to me. For several years I have been dealing with the possibilities of reducing climate warming by agriculture. Increasing the acreage of agricultural crops (including catch crops) increases the sequestration of carbon dioxide from the atmosphere. However, a significant proportion of the absorbed carbon returns back to the atmosphere as a result of the transpiration process. Therefore, the key to sustained success in retaining carbon is building a 'carbon bank' in the soil - retaining it as long as possible in the soil. This article meets such expectations - it arouses my interest. The obtained research results show that the lowest fertility of C was shown by arable land with corn and cotton (commodity plants of global importance), while the highest fertility - fallow land (5-10 years fallow). Another valuable conclusion from the research is the fact that its fraction (in this case the NOM fraction) is of great importance in retaining carbon in the soil. Such research results do not mean, however, that we will start massively converting arable land into fallow land in the world to improve the carbon balance in the soil - in jest. However, they show the direction in which to follow in crop management - sustainable agriculture. The intensification of crops favors unfavorable relations of the C content in the soil. My research shows that the cultivation of catch crops (as an additional plant cover during the growing season) positively influences the greater sequestration of carbon from the atmosphere and its subsequent retention in the soil.
The content of the article is generally well written. I only miss the research hypothesis clearly formulated in the "Introduction" - what assumptions did the authors adopt before starting this research? And have the assumptions made in the course of the research been confirmed or perhaps contradicted? These questions should be answered in "Conclusions". I believe that there is an urgent need for further similar research on the example of other plants and in other climate and soil zones in the world. Only then can far-reaching guidelines for managing this process be drawn.
This article is a valuable guide in this direction. After supplementing it with the suggested issues, it may be published in Carbon (MDPI). In the article, I also notice inconsistencies in the notation of SI units or notation errors, as well as other technical errors. However, I evaluate the article mainly in terms of its content - and this assessment is positive.
Author Response
Response to Reviewer 3
Comments and Suggestions for Authors
The subject of the article submitted for review is very close to me. For several years I have been dealing with the possibilities of reducing climate warming by agriculture. Increasing the acreage of agricultural crops (including catch crops) increases the sequestration of carbon dioxide from the atmosphere. However, a significant proportion of the absorbed carbon returns back to the atmosphere as a result of the transpiration process. Therefore, the key to sustained success in retaining carbon is building a 'carbon bank' in the soil - retaining it as long as possible in the soil. This article meets such expectations - it arouses my interest. The obtained research results show that the lowest fertility of C was shown by arable land with corn and cotton (commodity plants of global importance), while the highest fertility - fallow land (5-10 years fallow). Another valuable conclusion from the research is the fact that its fraction (in this case the NOM fraction) is of great importance in retaining carbon in the soil. Such research results do not mean, however, that we will start massively converting arable land into fallow land in the world to improve the carbon balance in the soil - in jest. However, they show the direction in which to follow in crop management - sustainable agriculture. The intensification of crops favors unfavorable relations of the C content in the soil. My research shows that the cultivation of catch crops (as an additional plant cover during the growing season) positively influences the greater sequestration of carbon from the atmosphere and its subsequent retention in the soil.
The content of the article is generally well written. I only miss the research hypothesis clearly formulated in the "Introduction" - what assumptions did the authors adopt before starting this research? And have the assumptions made in the course of the research been confirmed or perhaps contradicted? These questions should be answered in "Conclusions". I believe that there is an urgent need for further similar research on the example of other plants and in other climate and soil zones in the world. Only then can far-reaching guidelines for managing this process be drawn.
This article is a valuable guide in this direction. After supplementing it with the suggested issues, it may be published in Carbon (MDPI). In the article, I also notice inconsistencies in the notation of SI units or notation errors, as well as other technical errors. However, I evaluate the article mainly in terms of its content - and this assessment is positive.
Response 1
We thank the reviewer for their review and support of the Importance of this work.
Regarding the hypothesis and our assumption in conducting the study, we elaborated in the last portion of the introduction and concluded on our hypothesis in the conclusion of the reviewed manuscript.
English grammatical and spelling as well as consistency in the formatting were addressed and the reviewed manuscript was proof-red by native English expert.

Round 2
Reviewer 1 Report
The author has made substantial revision, and I suggest accept it.
Reviewer 2 Report
The MS has been greatly improved through the authors' revision. However, the format of references is still not correct. The authors should use standard abbr. of journal names in references. I encourage the authors correct the format in proofreading phase.